# SEQUENCE MODELLING WITH AUTO-ADDRESSING AND RECURRENT MEMORY INTEGRATING NETWORKS

## ABSTRACT

Processing sequential data with long term dependencies and learn complex transitions are two major challenges in many deep learning applications. In this paper, we introduce a novel architecture, the Auto-addressing and Recurrent Memory Integrating Network (ARMIN) to address these issues. The ARMIN explicitly stores previous hidden states and recurrently integrate useful past states into current time-step by an efficient memory addressing mechanism. Compared to existing memory networks, the ARMIN is more light-weight and inference-time efficient. Our network can be trained on small slices of long sequential data, and thus, can boost its training speed. Experiments on various tasks demonstrate the efficiency of the ARMIN architecture. Codes and models will be available.

## 1 INTRODUCTION

Recurrent neural networks, such as the Long Short-Term Memory (LSTM) (Hochreiter & Schmidhuber, 1997) and Gated Recurrent Unit (GRU) (Cho et al., 2014) have shown promising performance for processing sequential data. However, it's known that RNNs suffer from gradient vanishing problem. Moreover, as pointed out by Rae et al. (2016), the number of parameters grows proportionally to the square of the size of the hidden units, which carry the historical information. Recent memory-based approaches exhibit potential to address these issues, by decoupling memory capacity from model parameters, and backpropagating the gradients through the memory.

Neural Turing Machine (NTM) (Graves et al., 2014) first emerged as a recurrent model that incorporates external memory abilities. NTM maintains a memory matrix, and at every time-step, the network reads and writes (with erasing) to the memory matrix using certain soft-attentional mechanism, controlled by an LSTM that produces read and write vectors. NTM and its successor, the Differentiable Neural Computer(Graves et al., 2016), have shown success on some algorithmic tasks such as copying, priority sorting and some real-world tasks such as question answering. But one limitation of the NTM is that due to its smooth read and write mechanism, NTM has to do propagations on the entire memory, usually causing huge amount of memory consumption. To this end, Rae et al. (2016) proposes the Sparse Access Memory(SAM) network, by thresholding memory modifications to a sparse subset, *i.e.* all read and write operations are limited to several memory words. This allows memory and time efficient propagations while maintaining NTM's performance. However, these external memory models have relatively complicated memory addressing mechanisms, making them suitable for only large-scale memory. Moreover, the RNN in these models plays a simple role of being a controller, but the gradient vanishing problem of RNN itself is not given attention. In contrast to these networks, our network uses a light-weight read mechanism to address a small external memory, and relieves the gradient vanishing problem by directly concatenating memory word to output.

Inspired by prior memory models, efforts have been made to build a bridge between simple RNNs and complicated memory models. Kurach et al. (2015) propose the Neural Random-access Machines (NRAM) that can manipulate and dereference pointers to an external variable-size random-access memory. Danihelka et al. (2016) improve LSTM with ideas from Holographic Reduced Representations (Plate, 2003) that enables key-value storage of data. Grave et al. (2016) propose a method of augmenting LSTM by storing previous (hidden state, input word) pairs in memory and using the current hidden state as a query vector to recover historical input words. This method requires no backpropagation through memory and is well-suited to word-level language tasks. Grefenstette

et al. (2015); Dyer et al. (2015); Joulin & Mikolov (2015) augment RNNs with a stack structure that works as a natural parsing tool, and use them to process algorithmic and nature language processing (NLP) tasks; nonetheless, the speed of stack-augmented RNNs is rather slow due to multiple push-pop operations at every time-step. Ke et al. (2018) propose the Sparse Attentive Backtracking (SAB) architecture, which recalls a few past hidden sates at every time-step and do "mental" backpropagations to the nearby hidden states with respect to the past hidden states. Gulcehre et al. (2017) propose the TARDIS network, which is an LSTM-resembled RNN that directly stores a fixed number of previous hidden states with learned key parameters for memory addressing. At every time-step, the network reads out one historical state $r_t$ from the memory, and uses it, along with the input $x_t$ and last hidden state $h_{t-1}$, to produce new hidden state $h_t$, then overwrites $h_t$ to the location of $r_t$. TARDIS optionally uses the gumbel-softmax estimator(Maddison et al., 2016; Jang et al., 2016) to sample the location of $r_t$ (we will explain this in section 2.1), so the whole network is differentiable. However, the TARDIS still involves some hand-crafted memory addressing method and its addressing mechanism requires a considerable amount of memory and time consumption.

Inspired by the TARDIS, we introduce the ARMIN architecture, a more light-weight and memory-based architecture with a very simple memory addressing mechanism and in-depth modification of the existing LSTM structure. Concretely, our contributions are as follows:

• We distill previous memory addressing methods and propose a simple yet effective memory addressing mechanism for the external memory, namely Auto-addressing, by encoding the information for memory addressing directly via the inputs $x_t$ and the hidden states $h_{t-1}$.

• We propose a novel recurrent cell that combines the gating advantages of LSTM and allows direct gradient backpropagation from output to memory. With only a single cell, it learns better representations than many hierarchical RNN structures.

• We show in our ablation study that the ARMIN is robust to small iteration lengths when training long sequential data, which enables training with large batch sizes and boost its training speed. We further demonstrate in char-level language modelling task that the ARMIN obtain $40\%$ training speed gain than LSTM while keeping similar performance and memory consumption.

• We demonstrate competitive results on various tasks while keeping efficient time and memory consumption during training and inference time.

## 2    BACKGROUND

### 2.1    GUMBEL-SOFTMAX ESTIMATOR

Categorical distribution is a natural choice for representing discrete structure in the world; however, it's rarely used in neural networks due to its inability to backpropagate through samples(Jang et al., 2016). To this end, Maddison et al. (2016) and Jang et al. (2016) propose a continuous relaxation of categorical distribution and the corresponding gumbel-softmax gradient estimator that replaces the non-differentiable sample from a categorical distribution with a differentiable sample. Specifically, given a probability distribution $\boldsymbol{p} = (\pi_1, \pi_2, ..., \pi_k)$ over $k$ categories, the gumbel-softmax estimator produces an one-hot sample vector $\boldsymbol{y}$ with its $i$-th element calculated as follows:

$$y_i = \frac{\exp((\log(\pi_i) + g_i)/\tau)}{\sum_{j=1}^{k} \exp((\log(\pi_j) + g_j)/\tau)} \quad \text{for } i = 1, 2, ..., k, \tag{1}$$

where $g_1, ..., g_k$ are i.i.d samples drawn from Gumbel distribution(Gumbel, 1954):

$$g_i = -\log(-\log(u_i)), u_i \sim \text{Uniform}(0, 1), \tag{2}$$

and $\tau$ is the *temperature* parameter. In practice, we usually start at a high temperature and anneal to a small but non-zero temperature (Jang et al., 2016).

## 3    AUTO-ADDRESSING AND RECURRENT MEMORY INTEGRATING NETWORK

In this section, we describe the structure of the ARMIN network as shown in Figure 1. It consists of a recurrent cell and a small external memory that stores historical hidden states. While processing

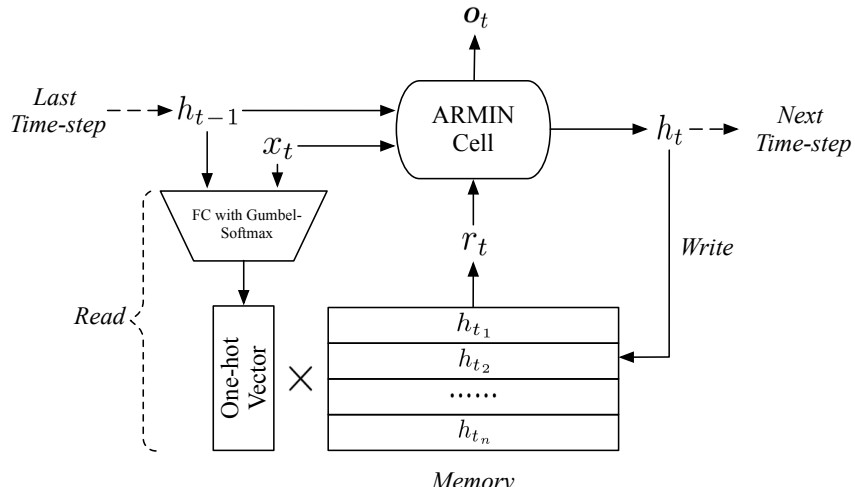

Figure 1: The ARMIN structure. At each time-step, the ARMIN performs read operation, cell processing and write operation in chronological order: **(a)** It reads out a historical hidden state $r_t$ from memory with an one-hot read vector produced via passing $x_t$ and $h_{t-1}$ to a fully connected layer followed by a gumbel-softmax function. **(b)** The ARMIN cell receives $x_t$, $h_{t-1}$ and $r_t$ as inputs and outputs $o_t$ and $h_t$. $o_t$ is passed to output layers, and $h_t$ is passed to next time-step. **(c)** $h_t$ is written to the previous location of $r_t$.

sequential data, the ARMIN performs reading from memory, cell processing and writing to memory operations in chronological order during each time-step. In the following subsections, we first explain the structure of the recurrent cell, and then discuss the read and write operations.

### 3.1 THE RECURRENT CELL OF ARMIN

Inspired by classical LSTM structure (please refer to APPENDIX A for details in RNN and LSTM structures), we propose a novel memory-augmented recurrent cell structure, namely the ARMIN cell. At every time-step, it takes in an input $x_t$, the last hidden state $h_{t-1}$ and a recovered historical hidden state $r_t$ chosen by a read operation, and produces an output vector $o_t$ and the new hidden state $h_t$. The computation process is as follows:

$$\left\{ \begin{matrix} g_t^h \\ g_t^r \end{matrix} \right\} = \left\{ \begin{matrix} \sigma \\ \sigma \end{matrix} \right\} W_{ig}[\, x_t, h_{t-1}, r_t \,] + b_{ig} \,, \tag{3}$$

$$h_{t-1}^g = g_t^h \circ h_{t-1} \,, \tag{4}$$

$$r_t^g = g_t^r \circ r_t \,, \tag{5}$$

$$\left\{ \begin{matrix} i_t \\ f_t \\ g_t \\ o_t^h \\ o_t^r \end{matrix} \right\} = \left\{ \begin{matrix} \sigma \\ \sigma \\ \tanh \\ \sigma \\ \sigma \end{matrix} \right\} W_{go}[\, x_t, h_{t-1}^g, r_t^g \,] + b_{go} \,, \tag{6}$$

$$h_t = f_t \circ h_{t-1} + i_t \circ g_t \,, \tag{7}$$

$$o_t = [\, o_t^h \circ \tanh(h_t) \,, o_t^r \circ \tanh(r_t) \,] \,. \tag{8}$$

where $h_{t-1}, h_t \in \mathbb{R}^{d_h}, r_t \in \mathbb{R}^{d_r}, x_t \in \mathbb{R}^{d_i}$, and $W_{ig} \in \mathbb{R}^{2d_h \times (d_i + d_h + d_r)}, W_{go} \in \mathbb{R}^{(4d_h + d_r) \times (d_i + d_h + d_r)}, b_{ig} \in \mathbb{R}^{d_h + d_r}, b_{go} \in \mathbb{R}^{4d_h + d_r}, o_t \in \mathbb{R}^{d_h + d_r}$. We refer to $d_h$ as the *hidden size* of the recurrent cell of ARMIN. Usually we have $d_h = d_r$ and allocate equal number of weight parameters for $h_t$ and $r_t$.

In equation $3 \sim 5$, two gates are calculated to control the information flow for $h_{t-1}$ and $r_t$ respectively, generating gated hidden state $\boldsymbol{h}_{t-1}^g$ and historical state $\boldsymbol{r}_t^g$; using this method, we can filter out the irrelevant information for the current time-step. Then as shown in equation 6, we compute the input gate $\boldsymbol{i}_t$, forget gate $\boldsymbol{f}_t$, cell state $\boldsymbol{g}_t$ and output gate $\boldsymbol{o}_t^h$ for the new hidden state just like in classical LSTM structure. Additionally, an output gate $\boldsymbol{o}_t^r$ for historical state $\boldsymbol{r}_t$ is computed. Next in equation 7, we compute new hidden state $\boldsymbol{h}_t$ that is the sum of $\boldsymbol{h}_{t-1}$ and cell state $\boldsymbol{g}_t$, leveraged by forget gate $\boldsymbol{f}_t$ and input gate $\boldsymbol{i}_t$. Finally in equation 8, we calculate the output of this time-step, which is the concatenation of the gated contents from $\boldsymbol{h}_t$ and $\boldsymbol{r}_t$. Using this method, we can *selectively backpropagate gradients* from the output to the historical state $\boldsymbol{r}_t$. We involve $\boldsymbol{r}_t$ (or $\boldsymbol{r}_t^g$) in all gates' computations.

Intuitively, the $\boldsymbol{h}_{t-1}$ acts as the old working memory, and $\boldsymbol{r}_t$ is treated as the long-term memory. The cell processes them with the input $\boldsymbol{x}_t$ to generate the new working memory $\boldsymbol{h}_t$ and the output $\boldsymbol{o}_t$. More specifically, each $\boldsymbol{r}_t$ is a summary of historical hidden states selected by auto-addressing mechanism. The ARMIN cell learns to recurrently integrate the summary of long-term information from $\boldsymbol{r}_t$ into the working memory $\boldsymbol{h}_t$. We will demonstrate the efficiency of recurrent memory integration in language modelling tasks.

Unlike the LSTM that passes a tuple of $(\boldsymbol{h}_t, \boldsymbol{c}_t)$ to the next recurrence and outputs $\boldsymbol{h}_t$ at the same time, the ARMIN only passes one hidden state $h_t$ and separately uses $\boldsymbol{o}_t$ as output, which has 2 benefits: one is that it takes lower memory consumption to store one hidden state rather than two, and the other is we can decouple the information needed for the output $\boldsymbol{o}_t$, from the information (*i.e.* $\boldsymbol{h}_t$) that is needed for memorizing and being used at later time-steps. Furthermore, we list 3 important differences between the ARMIN cell and TARDIS cell, please refer to APPENDIX B.1.

### 3.2 READ OPERATION WITH AUTO-ADDRESSING

The ARMIN maintains a memory matrix $\boldsymbol{M} \in \mathbb{R}^{n_{mem} \times d_h}$, where the constant $n_{mem}$ denotes the number of memory slots. At each recurrence, the ARMIN chooses a historical state $r_t$ from memory according to the information in $\boldsymbol{x}_t$ and $\boldsymbol{h}_{t-1}$, which is formulated as follows:

$$\boldsymbol{s}_t = \text{gumbel-softmax}(\boldsymbol{W}_s[\,\boldsymbol{x}_t\,,\boldsymbol{h}_{t-1}\,] \,+\, \boldsymbol{b}_s)\,, \tag{9}$$

$$\boldsymbol{r}_t = \sum_{i=0}^{n-1} s_t(i)\boldsymbol{M}(i,:)\,. \tag{10}$$

where $\boldsymbol{W}_s \in \mathbb{R}^{n_{mem} \times (d_i+d_h)}$, $\boldsymbol{b}_s \in \mathbb{R}^{n_{mem}}$, $\boldsymbol{s}_t$ is a one-hot vector sampled by gumbel-softmax function, $s_t(i)$ denotes the $i$-th element of $\boldsymbol{s}_t$, $\boldsymbol{M}(i,:)$ denotes the $i$-th row of $\boldsymbol{M}$.

As opposed to previous memory networks such as Santoro et al. (2016) and Gulcehre et al. (2017), we don't use any extra *usage* vectors and learnable key parameters that manually encode memory addressing information to assist memory addressing. As our ablation study (Appendix E) shows, the hidden state $\boldsymbol{h}_{t-1}$ is sufficient to encode the historical memory accessing information. Furthermore, the TARDIS addressing mechanism requires concatenating $\boldsymbol{x}_t$ and $\boldsymbol{h}_t$ to every memory cell, which usually causes large extra memory and time consumption in practical scenes. Please refer to APPENDIX B.2 for more precise comparison regarding addressing mechanism of the ARMIN and TARDIS networks.

### 3.3 WRITE OPERATION

After the reading and cell processing stages, the ARMIN writes the new hidden state $\boldsymbol{h}_t$ to the memory $\boldsymbol{M}$. Following the TARDIS network (Gulcehre et al., 2017), we simply overwrite $\boldsymbol{h}_t$ to the memory slot where we just read out the $\boldsymbol{r}_t$ (for conditions where $d_h$ is not equal to $d_r$, we first use a linear layer to transform $\boldsymbol{h}_t$ from $d_h$ dimension to $d_r$ dimension). But at the initial time-steps, we write the hidden states to the empty memory slots, until all empty slots are filled with historical states. In this way, we can maximally preserve useful historical information, because the ARMIN cell can learn to copy the useful information from $\boldsymbol{r}_t$ to $\boldsymbol{h}_t$, and then write $\boldsymbol{h}_t$ to the previous location of $\boldsymbol{r}_t$. The read/write mechanism can be viewed as a form of a skip-connection of hidden states. By training the whole network, the ARMIN can easily learn long-term dependencies via direct access to historical hidden states. From this point of view, our network is similar to the Skip RNN proposed

by Campos Camunez et al. (2018), which differs from our network in that they use a binary state update gate to select whether the state of the RNN will be updated or copied from the previous time-step.

# 4 EXPERIMENTS

We evaluate our model on algorithmic tasks, character-level language modelling task and temporal action detection and proposal task. We compare our network with NTM (with an LSTM controller), TARDIS and vanilla LSTM networks. We also directly compare the auto-addressing against the TARDIS addressing mechanism, by replacing auto-addressing with the TARDIS addressing method in ARMIN. We refer to this network as ARMIN+TARDIS-addr in our experiments. Please refer to APPENDIX C for some universal setups used in all our experiments below.

## 4.1 ALGORITHMIC TASKS

Along with the NTM, Graves et al. (2014) introduced a set of synthetic algorithm tasks which consists of copy, repeat copy, associative recall, N-gram and priority sort tasks. Here we use four out of five of these tasks (excluding the N-gram task) to examine if our network can choose correct time-steps from the past and effectively make use of them. Please refer to APPENDIX C.1 for more explanations and hyperparameter setups with respect to these tasks. Note that we have a strong LSTM baseline with about 4 times larger parameter count than memory networks, and the training of our copy and priority sort tasks are more challenging than the original tasks in Graves et al. (2014), for having more than 2 times longer input and target sequences. The models are optimized to minimize the averaged binary cross-entropy loss on the target multi-bit binary sequences. Following Gulcehre et al. (2017); Campos Camunez et al. (2018), in all tasks, we consider a task solved if the validation loss is at least two orders of magnitude below the initial cost which is around 0.70, $i.e.$ the validation loss (the validation set is generated randomly) converges to less than 0.01 (with less than 30% in 10 consecutive validation of sharp losses that are higher than 0.01 ). In our evaluation, we are interested in if a model can successfully solve the task in 100k iterations and the elapsed training time and iterations till the model succeeds on the task. For all networks we ensure the similar total parameter counts. All important hyperparameters are shown in APPENDIX C.1. We run all experiments with batch size of 1 under 2.8 GHz Intel Core i7 CPU and report the wall clock time and the number of iterations for solved tasks of each model. For models that fail to converge to less than 0.01 loss in 100k iterations, we report the average loss of the final 10 validations (denoted in underlines) and the elapsed training time for 100k iterations. The results are shown in Table 1. The training curves for each task are shown in APPENDIX D.

The results show that the ARMIN can solve 3 out of 4 tasks in a fast converge speed. By comparing the ARMIN with NTM, we observe that the NTM is able to almost solve all of the tasks, but with a much slower training speed compared to the ARMIN, for example, the training time of NTM is 5 times larger than ARMIN to solve the copy task. In fact, the training speed of ARMIN for each iteration is about $3 \sim 4$ times faster than the NTM's speed. We observe that the TARDIS and ARMIN+TARDIS-addr fail to solve 3 out 4 tasks in the given 100k iterations. By comparing them with the ARMIN, we confirm the efficiency of the auto-addressing of ARMIN, for bringing a fast

Table 1: The average elapsed time(min) and iterations(k) of different networks till task solved in given 100k iterations. The wall clock training time for 100k iterations and the average loss of final 10 validations (denoted in underlines) are shown for the unsolved tasks.

|  | Copy | | Repeat Copy | | Associative Recall | | Priority Sort | |
|---|---|---|---|---|---|---|---|---|
|  | time | iter. | time | iter. | time | iter. | time | iter. |
| LSTM | 66 | 0.359 | 45 | 0.016 | 34 | 0.325 | 32 | 37.0 |
| NTM | 32 | 12.4 | 171 | 0.014 | 22 | 19.6 | 360 | 0.012 |
| TARDIS | 157 | 0.451 | 97 | 0.166 | 66 | 0.330 | 133 | 62.6 |
| ARMIN+TARDIS-addr | 152 | 0.410 | 95 | 0.297 | 64 | 0.336 | 151 | 71.4 |
| ARMIN | 6 | 7.6 | 36 | 67.8 | 33 | 0.052 | 37 | 33.4 |

Table 2: Bits-per-character on Penn Treebank and enwik8 test set.

| Model | PTB BPC | PTB Params | enwik8 BPC | enwik8 Params |
|---|---|---|---|---|
| LSTM | 1.36 | – | 1.45 | – |
| LSTM+Zoneout (Krueger et al., 2016) | 1.27 | – | – | – |
| LSTM+Layer Norm(1000 units)[1] | 1.267 | 4.26M | – | – |
| LSTM+Layer Norm+Zoneout(1024 units)[2] | 1.24 | 4.79M | – | – |
| HM-LSTM+Layer Norm (Chung et al., 2016) | 1.24 | – | 1.32 | 35M |
| HyperLSTM+Layer Norm(Ha et al., 2016) | 1.219 | 14.41M | 1.340 | 26.5M |
| NASCell (Zoph & Le, 2016) | 1.214 | 16.28M | – | – |
| IndRNN (21 layers) (Li et al., 2018) | 1.21 | – | – | – |
| Recurrent Highway Hypernetwork (Suarez, 2017) | 1.19 | 15.5M | – | – |
| Fast-Slow LSTM (Mujika et al., 2017) | 1.190 | 7.2M | 1.277 | 27M |
| NTM(800 units, $n_{head}=1$, $p_{dropout}$=0.6) | 1.535 | 8.28M | – | – |
| TARDIS(paper) | 1.25 | – | – | – |
| TARDIS(1000 units,$p_{dropout}$=0.6) | 1.268 | 9.2M | – | – |
| ARMIN+TARDIS-addr(800 units,$p_{dropout}$=0.6) | 1.223 | 10.2M | – | – |
| ARMIN (500 units, $p_{dropout}$=0.4) (Ours) | 1.236 | 4.03M | – | – |
| ARMIN (800 units, $p_{dropout}$=0.6) (Ours) | 1.202 | 9.8M | – | – |
| ARMIN (800 units, 2 layer) (Ours) | – | – | 1.331 | 21.6M |

converge speed and short training time. We also find that the LSTM, when given enough parameter counts (4 times larger than memory networks), is able to solve the priority sort task quickly and converge to a better performance than TARDIS and ARMIN+TARDIS in other tasks. We believe there are 2 possible reasons for the poor performance of the TARDIS addressing mechanism: **a)** The tanh activation (see equation 21 in APPENDIX B.2) might cause the gradient vanishing problem when it cooperates with the gumbel-softmax function. **b)** The multiple inputs of $\boldsymbol{x}_t, \boldsymbol{h}_t, \boldsymbol{M}_t[i], \boldsymbol{u}_t$ in equation 21 cause the network hard to initialize to keep equal input and output variances for the linear layer and might also cause gradient dispersion. The auto-addressing has neither of these two issues and can better integrately backpropogate gradients, therefore it can perform well and quickly on these tasks. However, we would like to point out 2 limitations of the auto-addressing: **a)** The associative recall task shows the auto-addressing might have problems in establishing indirect references among memory data. **b)** The auto-addressing is relatively hard to generalize to unseen longer sequence in tasks such as copy and repeat copy due to its simplicity, whereas the NTM is able to do that. We leave these 2 issues for future work.

## 4.2 CHARACTER-LEVEL LANGUAGE MODELLING

The character-level language modelling task consists of predicting the probability distribution of the next character given all the previous ones, and we use the widely used Penn Treebank and Hutter Prize Wikipedia (also known as $enwik8$) datasets for this task. We compare our network with vanilla LSTMs and other memory networks on Penn Treebank, and compare with some popular RNN variants on both datasets. We ensure similar parameter counts and hyperparameters for all memory networks. Following prior works, we apply truncated backpropagation through time(TBPTT) (Rumelhart et al., 1986; Elman, 1990) to approximate the gradients: at each iteration, the network predict the next 150 characters, and the hidden state $\boldsymbol{h}_t$ and memory state $\boldsymbol{M}$ are passed to the next iteration. The gradients are truncated between different iterations. Please refer to APPENDIX C.2 for more experimental details.

The results are shown in table 2. On Penn Treebank dataset, our best performing network achieves competitive 1.202 BPC on Penn Treebank dataset, which is the best single cell performance that we are aware of, with relatively small parameter count. By comparing TARDIS and ARMIN+TARDIS-

---

[1]As implemented in Ha et al. (2016).

[2]Our implementation. We don't see performance growth when we further increase the hidden size.

addr, we observe about 4 points of improvement, which shows the efficiency of the ARMIN cell. By comparing ARMIN+TARDIS-addr, we observe a further improvement of around 2 points, which shows the efficiency of the auto-addressing mechanism. We also find that the same architecture of NTM that performs well in algorithmic tasks fails to converge to a lower BPC than vanilla LSTM, even with various regularizations we add to NTM.

The results on Penn Treebank show our single ARMIN cell learns better representations than many hierarchical RNN structures, such as the HM-LSTM, 2-Layer HyperLSTM and 21 layer IndRNN. Our network is outperformed by Recurrent Highway Hypernetwork and Fast-slow LSTM which are both multi-scale and deep transition RNNs and are state-of-the-art results without any auxiliary technics on this dataset. By saying "better representations", we refer to the concatenation of the gated contents from $h_t$ and $r_t$ as in equation 8. For example, if we remove the gated contents of $r_t$ from $o_t$, the ARMIN undergoes a BPC performance drop from 1.202 to 1.220, which is still better than the best performing BPC of 1.24 of the LSTM. We believe the rest of the performance gain comes from the recurrent memory integration of the ARMIN cell, which also favors a paradigm of deep transition, as is shown by the success of the deep transition RNNs on this task. For more ablation study regarding the auto-addressing mechanism and the ARMIN cell please refer to APPENDIX E (where we show the robustness of ARMIN to small TBPTT length).

The result on enwik8 demonstrates a simple 2-layer ARMIN can achieve competitive BPC performance of 1.33, with less parameter count compared to the HyperLSTM and HM-LSTM. By constructing deeper ARMIN network or even combining with other multi-scale and hierarchical RNN architectures, we believe the performance can be further improved.

### 4.3    Temporal action detection and proposals on THUMOS' 14

In this subsection, we evaluate the ARMIN on a more complicated real-world task concerning video analysis, *i.e.* the temporal action detection and proposals, which consists of taking an input video and producing a set of temporal intervals that are likely to contain human actions (Buch et al., 2017). We use the Single-Stream Temporal Action Proposal(SST) (Buch et al., 2017) as our framework, and evaluate the model on the THUMOS' 14 dataset (Jiang et al., 2014), which contains 20+ hours of video with 200 train/validation and 213 test untrimmed video sequences. The SST consists of a GRU and a fully connected output layer. In our experiment, we replace the GRU encoder in SST with a single ARMIN cell, and we call the modified network MA-SST. For fairness, we also implement a modified SST with LSTM encoder and compare them with the original SST in Buch et al. (2017). Please refer to APPENDIX C.3 for more experimental details.

Table 3: Comparison of proposal generation performance in terms of recall at 1000 proposals. We choose tIoU=0.6 and 0.8 to compare for consistency with Buch et al. (2017). Our MA-SST achieves highest recall at tIoU=0.8.

| Model | tIoU=0.6 | tIoU=0.8 |
|---|---|---|
| SST (Paper) | **0.920** | 0.672 |
| SST (We implement) | 0.897 | 0.732 |
| MA-SST(Ours) | 0.911 | **0.759** |

Table 4: Comparison for performance and cost of different setups on Penn Treebank dataset.

| Model | ARMIN | | LSTM | |
|---|---|---|---|---|
| Setup | 1 | 2 | 1 | 2 |
| Hidden size | 500 | 550 | 1k | 1k |
| $n_{param}$(M) | 4.02 | 4.81 | 4.79 | 4.79 |
| $n_{mem}$ | 5 | 10 | – | – |
| $T_{trunc}$ | 50 | 50 | 100 | 150 |
| batch size | 384 | 300 | 128 | 128 |
| Memory(GB) | 3.49 | 3.56 | 2.36 | 3.27 |
| Speed (chars/s) | 98k | 75k | 71k | 70k |
| BPC | 1.238 | 1.226 | 1.27 | 1.24 |

The average recall under different proposal numbers and tIoUs[3] are depicted in figure 2. MA-SST outperforms the original SST and our SST simultaneously. To the best of our knowledge, MA-SST achieves highest performance of recall at around tIoU=0.8 (see table 3), even if other competitive non-RNN networks (Escorcia et al., 2016; Caba Heilbron et al., 2016; Gao et al., 2017; Guo et al., 2018) are taken into account. The MA-SST tends to precisely generate proposals that has a high overlapping area with the ground truth action intervals, leading to high recall performance at high

---

[3]tIoU denotes temporal Intersection over Union between a proposal and its maximally overlapped ground truth action. Applying bigger tIoU threshold make the generated proposals higher in quality but less in quantity.

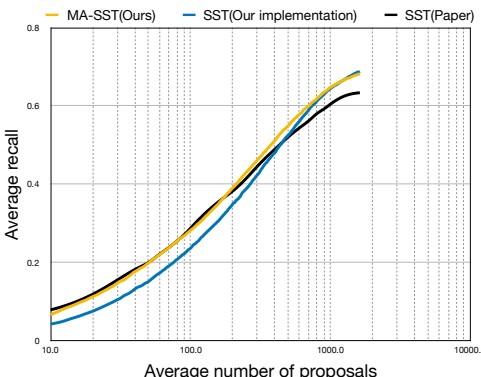 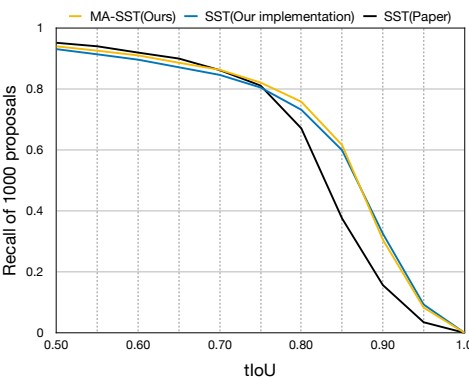

Figure 2: Comparison of proposal generation performance. For data generation, we use the code offered in Buch et al. (2017) for consistency. (Left) The average number of proposals v.s. average recall for tIoU $\geq 0.5$. The MA-SST outperforms our SST when average number of proposals is less than 500, and outperforms the original SST when average number of proposals is more than 500. (Right) Different tIoU v.s. recall of 1000 average proposals. The MA-SST strongly outperforms the original SST and outperform our SST by a maximal margin of 2.7%.

tIoUs. We believe the MA-SST has better performance than the SST mainly because of the direct access to historical states from which it can better locate the starting frame of an action.

## 5 TOWARDS MORE LIGHT-WEIGHT RECURRENT MEMORY NETWORKS

It's known that the speed and computer memory consumption have important influences on the practicability of an RNN, especially when external memory is involved. We have shown in some of the algorithmic tasks that the ARMIN can be trained 3 4 times faster than the NTM in terms of wall clock time. Next we conduct a more thorough comparison experiment among the vanilla LSTM and memory networks under different hidden sizes. The experiment is based on the character-level language modelling task on Penn Treebank dataset, without the loss of generality. For a fair comparison, we implement all networks with near-optimal implementations using Pytorch, an efficient deep-learning framework. We use the aforementioned experiment setup, and keep all memory matrices the same size, *i.e.* $20 \times d_h$. We run the experiment using single-precision floating point calculations under a Titan XP GPU that has 12GB memory space. The results are depicted in figure 3. From the results we observe 2 important phenomena: **a)** The ARMIN consistently outperform other memory networks shown in the graph in terms of running speed both at training and inference stages, and the main contribution to this comes from the simple auto-addressing mechanism of the ARMIN. Moreover, at inference stage, we can replace the memory matrix with a list of discrete memory slots, and update memory by simply replacing the old hidden states with the new ones, furthermore, we can replace the gumbel-softmax function with argmax. Using these methods, ARMIN's inference speed obtains significant improvement than in training stage. **b)** The ARMIN outperforms TARDIS and ARMIN+TARDIS-addr in terms of training memory consumption, mainly because the TARDIS addressing mechanism requires concatenating $x_t$ and $h_{t-1}$ to every memory cell for parallelization. At inference stage, we observe TARDIS have smaller memory consumption than ARMIN and ARMIN+TARDIS-addr, mainly because TARDIS has smaller parameter counts under the same hidden size. However, we would like to point out that under the same parameter counts the ARMIN usually outperforms TARDIS as we show in algorithm and language modelling tasks. We also observe the NTM has the largest memory consumption at inference time, mainly because its complex addressing mechanism.

We have shown in the ablation study (see APPENDIX E) that our network is robust to small iteration length when trained with TBPTT. We realize that we can use this feature to enable large batch size, and boost ARMIN's training speed while keeping better performance than LSTM. To validate this benefit, we conduct an experiment to compare the performance and cost of ARMIN against our best performing LSTM. We also provide the test result of the same LSTM under TBPTT length $T_{trunc} = 100$ for better comparing. The hyperparameter setups and results are shown in Table 4.

By comparing ARMIN setup 1 with LSTM setup 2, we observe that with only $6.7\%$ more memory consumption, we obtain $40\%$ training speed gain while keeping a slightly better BPC performance and $16\%$ less parameter count; by comparing ARMIN setup 2 with LSTM setup 2, we observe that with only $8.8\%$ more memory consumption, we obtain about 1.5 points of BPC performance gain and $7.1\%$ training speed gain under similar parameter count.

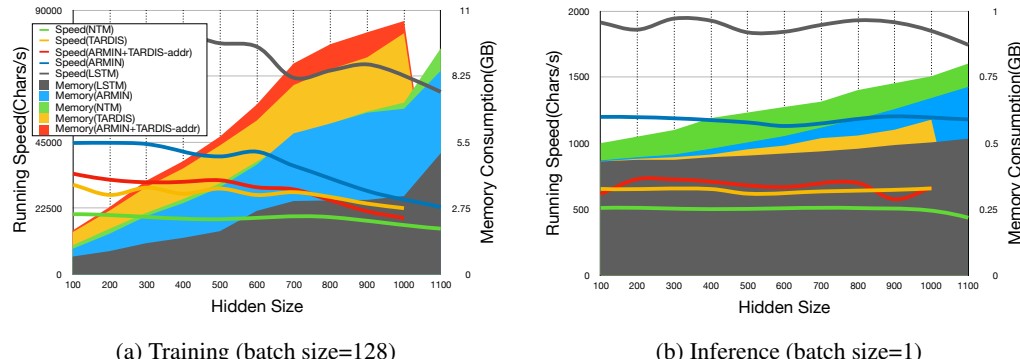

(a) Training (batch size=128)          (b) Inference (batch size=1)

Figure 3: The running speed and memory consumption at the training and inference stages ($n_{mem} = 20$). The solid blocks shows the memory consumption and the curves denote the running speed in characters/s. **(a)** shows the training stage, and **(b)** shows the inference stage (note that ARMIN+TARDIS-addr has basically the same memory consumption with ARMIN, so it's not shown in the graph).

## 6    CONCLUSION

In this paper, we have introduced the ARMIN, a light-weight and memory-augmented RNN architecture with a novel ARMIN cell. The ARMIN incorporates an efficient external memory with the light-weight auto-addressing mechanism. We demonstrate competitive performance of ARMIN in various tasks, and shows the generality of our model. Our ablation study suggests the efficacy of our external memory and addressing mechanism, and notably, our network is robust to short-length TBPTT which enables using large batch size to speed up the training and further increase performance on sequence modelling tasks. Further research may lead to the efficient hierarchical and multi-scale structures of the ARMIN and its successful applications in Seq2Seq models and complicated reasoning tasks.

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

## APPENDIX A    RNN AND LONG SHORT-TERM MEMORY

A recurrent neural network (RNN) is a class of neural network that recurrently processes a sequence of inputs $\{\boldsymbol{x}_1, \boldsymbol{x}_2, ..., \boldsymbol{x}_T\}$, and returns a sequence of outputs $\{\boldsymbol{y}_1, \boldsymbol{y}_2, ..., \boldsymbol{y}_T\}$ where $\boldsymbol{x}_t \in \mathbb{R}^{d_i}$, $\boldsymbol{y}_t \in \mathbb{R}^{d_o}$, and $d_i, d_o$ is the input size and output size, respectively. In a vanilla RNN, $\boldsymbol{y}_i$ is given by following equations:

$$\boldsymbol{h}_t = \sigma(\boldsymbol{U}\boldsymbol{x}_t + \boldsymbol{W}\boldsymbol{h}_{t-1} + \boldsymbol{b}) \,, \tag{11}$$

$$\boldsymbol{y}_t = \boldsymbol{V}\boldsymbol{h}_t + \boldsymbol{c} \,, \tag{12}$$

where the hidden state $\boldsymbol{h}_t \in \mathbb{R}^{d_h}$ is passed to the next recurrence. $d_h$ is the hidden size, and $\boldsymbol{U}, \boldsymbol{W}, \boldsymbol{V}, \boldsymbol{b}, \boldsymbol{c}$ are learnable parameters.

The LSTM (Hochreiter & Schmidhuber, 1997) is designed to overcome the gradient vanishing and exploding problem in vanilla RNN and learn long-term dependencies, by passing a tuple of $(\boldsymbol{h}_t, \boldsymbol{c}_t)$ to the next recurrence. Its computation process is defined as follows:

$$\begin{Bmatrix} \boldsymbol{i}_t \\ \boldsymbol{f}_t \\ \boldsymbol{g}_t \\ \boldsymbol{o}_t \end{Bmatrix} = \begin{Bmatrix} \sigma \\ \sigma \\ \tanh \\ \sigma \end{Bmatrix} \boldsymbol{W}_0[\,\boldsymbol{x}_t, \boldsymbol{h}_{t-1}\,] + \boldsymbol{b}_0 \,, \tag{13}$$

$$\boldsymbol{c}_t = \boldsymbol{f}_t \circ \boldsymbol{c}_{t-1} + \boldsymbol{i}_t \circ \boldsymbol{g}_t \,, \tag{14}$$

$$\boldsymbol{h}_t = \boldsymbol{o}_t \circ \tanh(\boldsymbol{c}_t), \tag{15}$$

where $\circ$ is the element-wise product, $\boldsymbol{W}_0 \in \mathbb{R}^{4d_h \times (d_i+d_h)}$, $\boldsymbol{b}_0 \in \mathbb{R}^{4d_h}$, $\boldsymbol{i}_t, \boldsymbol{f}_t, \boldsymbol{g}_t \, \boldsymbol{o}_t$ are the input gate, forget gate, cell state, output gate at time $t$, respectively, and they control the information flow in the LSTM.

## APPENDIX B    DIFFERENCES BETWEEN TARDIS AND ARMIN

The major differences between ARMIN and TARDIS networks are the recurrent cell computation and read operation. We show these differences respectively in subsections below:

### B.1    CELL COMPUTATIONS

The cell computation process of TARDIS is as follow:

$$\begin{Bmatrix} \boldsymbol{i}_t \\ \boldsymbol{f}_t \\ \boldsymbol{o}_t \end{Bmatrix} = \begin{Bmatrix} \sigma \\ \sigma \\ \sigma \end{Bmatrix} \boldsymbol{W}[\,\boldsymbol{x}_t, \boldsymbol{h}_{t-1}, \boldsymbol{r}_t\,] + \boldsymbol{b} \,, \tag{16}$$

$$\begin{Bmatrix} \alpha_t \\ \beta_t \end{Bmatrix} = \begin{Bmatrix} \text{gumbel-sigmoid} \\ \text{gumbel-sigmoid} \end{Bmatrix} \begin{Bmatrix} \boldsymbol{w}^{\alpha\top} \\ \boldsymbol{w}^{\beta\top} \end{Bmatrix} [\,\boldsymbol{x}_t, \boldsymbol{h}_{t-1}, \boldsymbol{r}_t\,] \,, \tag{17}$$

$$\boldsymbol{g}_t = \tanh(\boldsymbol{W}^g[\boldsymbol{x}_t, \alpha_t \boldsymbol{h}_{t-1}, \beta_t \boldsymbol{r}_t] + \boldsymbol{b}^g), \tag{18}$$

$$\boldsymbol{c}_t = \boldsymbol{f}_t \circ \boldsymbol{c}_{t-1} + \boldsymbol{i}_t \circ \boldsymbol{g}_t \,, \tag{19}$$

$$\boldsymbol{h}_t = \boldsymbol{o}_t \circ \tanh(\boldsymbol{c}_t) \,, \tag{20}$$

where $\boldsymbol{h}_{t-1}, \boldsymbol{h}_t, \boldsymbol{c}_t, \boldsymbol{i}_t, \boldsymbol{f}_t, \boldsymbol{o}_t, \boldsymbol{g}_t \in \mathbb{R}^{d_h}, \boldsymbol{r}_t \in \mathbb{R}^{d_r}, \boldsymbol{x}_t \in \mathbb{R}^{d_i}, \boldsymbol{w}^\alpha, \boldsymbol{w}^\beta \in \mathbb{R}^{d_i+d_h+d_r}$, and $\boldsymbol{W} \in \mathbb{R}^{3d_h \times (d_i+d_h+d_r)}, \boldsymbol{W}^g \in \mathbb{R}^{d_h \times (d_i+d_h+d_r)}, \boldsymbol{b} \in \mathbb{R}^{3d_h}, \boldsymbol{b}^g \in \mathbb{R}^{d_h}$. The output of TARDIS is the concatenation of $\boldsymbol{h}_t$ and $\boldsymbol{r}_t$.

There are 3 important points that distinguish the ARMIN cell from TARDIS cell, which are as follows:

- The TARDIS uses two binary scalar gates $\alpha_t, \beta_t$ to control the information flow in equation 17. The two scalar gates are activated using gumbel-sigmoid (which is similar to gumbel-softmax and the output is close to binary), whereas the ARMIN uses two soft vector gates to control the information flow, which is more flexible and enable larger model capacity. For example, in the char-level language modelling experiments, if we replace the sigmoid activations in equation 3 with the element-wise gumbel-sigmoid, the BPC performance immediately drops from 1.202 to 1.235.

- The TADIS apply information control only when calculating the cell state $g_t$, whereas the ARMIN apply information control immediately after the read operation. Our main motivation to this change is that if we don't filter out irrelevant information in $r_t$ and $h_t$ at first, these noisy information will cause the weight parameters hard to learn. Furthermore, if the read operation chooses the wrong past state, we can block the information flow for $r_t$ at the first place.

- The ARMIN has an extra gate $o_t^r$ to control the output from $r_t$ as in equation 8, which the TARDIS doesn't have. If we remove this gate and directly output $r_t$ like in TARDIS, we get a BPC performance drop from 1.202 to 1.219 in char-level language modelling task.

Finally, despite that the ARMIN has 3 more vector gates than the TARDIS, we have shown in the $p$MNIST and Penn Treebank experiments that the ARMIN can still outperform the TARDIS under similar parameter counts.

## B.2 READ OPERATION

The memory matrix $M_t$ of TARDIS has disjoint address section $A_t \in \mathbb{R}^{n_{mem} \times d_k}$ and content section $C_t \in \mathbb{R}^{n_{mem} \times d_r}$, $M_t = [A_t; C_t]$. The controller reads both the address and the content parts of the memory, but it will only write into the content section of the memory.

The TARDIS paper explored two ways to sample the read locations——using reinforce and using gumbel-softmax function, respectively. The gumbel-softmax way shows a better performance and its read operation can be formulated as follow:

$$\pi_t[i] = a^\top \tanh(W^\gamma[h_t, x_t, M_t[i], u_t] + b^\gamma), \tag{21}$$

$$s_t = \text{gumbel-softmax}(\pi_t). \tag{22}$$

where $\{a, W^\gamma, b^\gamma\}$ are learnable parameters (note that in our experiments, we always set the dimension of $a$ to about 1/4 of the hidden size) and $u_t$ is a hand-crafted usage vector which denotes the frequency of accesses to each cell in the memory.

There are two important differences with respect to TARDIS addressing mechanism and our auto-addressing mechanism:

- In practice, equation 21 requires concatenating $x_t$ and $h_t$ to every memory cells $M_t[i]$ for parallelization, which often causes significant extra memory and time consumption compared to the auto-addressing of ARMIN, as we show in the algorithmic tasks and char-level language modelling task.

- The auto-addressing omits the learnable address section $A_t$, the hand-crafted $u_t$ vector, the $\tanh$ activation and an auxiliary vector $a$ compared to TARDIS addressing mechanism. The TARDIS also requires manually subtracting 100 in unnormalized probability at last read location to avoid repeated read, whereas the auto-addressing doesn't need to. With these simplifications, we found the auto-addressing mechanism easier to train and it leads to a much faster converge speed, as we show in the algorithmic tasks. We also observe a BPC performance growth from 1.223 to 1.202 in char-level language modelling task when we switch from ARMIN+TARDIS-addr to ARMIN with auto-addressing.

## APPENDIX C EXPERIMENT SETUPS

We first describe some universal setups in all our experiments (unless we mention them again with different setup). Our model can be trained end-to-end, due to the utilization of gumbel-softmax function. As pointed out in Maddison et al. (2016), when the temperature $\tau \leq (n-1)^{-1}$, the

Table 5: Hyperparameters of the models for algorithmic tasks.

| Model | Hidden Size | memory Size | Param Count |
|---|---|---|---|
| LSTM | 300 | – | 376k |
| NTM | 120 | 128×20 | 88k |
| TARDIS | 120 | 50×32 | 90k |
| ARMIN+TARDIS | 100 | 50×32 | 91k |
| ARMIN | 100 | 50×32 | 90k |

Table 6: Hyperparameters of the algorithmic tasks.

| Task | Setup |
|---|---|
| Copy | copy length = 1~50 |
| Repeat Copy | copy length = 1~10 
 number of repeats = 1~10 |
| Associative Recall | item size = 3×6 
 item number = 2~6 |
| Priority Sort | Sort length = 40 |

density function of gumbel-softmax estimator is log-convex, which gives us a good guarantee for optimization. However, smaller $\tau$ will also bring higher variance of the gradients (Jang et al., 2016). So in all experiments, we initialize the reciprocal of $\tau$ as 1, and increase it by 1 after each epoch, until the reciprocal equals to $n - 1$. This scheme has shown better performance than other schemes.

We apply Layer Normalization(Ba et al., 2016) to all networks. In the ARMIN case, layer norm is applied to equation 3, 6 and 7, which we find important to regulate the hidden state when scale grows. To avoid overfitting, we apply dropout (Srivastava et al., 2014) to the input and output layers, and Zoneout(Krueger et al., 2016) is applied for recurrent connections. The networks are trained with Adam optimizer (Kingma & Ba, 2014). All weight matrices in RNNs are initialized to orthogonal matrices, and the bias of the forget gate $\boldsymbol{f}_t$ is initialized to 1. For all models using TARDIS addressing mechanism, we set the memory key size to about 1/5 of the memory content size.

## C.1 ALGORITHMIC TASKS

In all algorithmic experiments below, following Graves et al. (2014), we apply RMSProp optimizer (Tieleman & Hinton, 2012) with a momentum of 0.9. The learning rate is 0.0001 and gradient norms are clipped to 10.0. No regularization is used as the size of the networks is small. Following Graves et al. (2014), we have learned bias of initial hidden states and memory matrices for all memory networks. For TARDIS, ARMIN+TARDIS-addr and ARMIN we increase the reciprocal of temperature $\tau$ by 1 for every 200 iterations. The hyperparamters for models and tasks are shown in Table 5 and Table 6, respectively. We explain these tasks in detail in subsections below:

### C.1.1 COPY

The copy task tests whether a recurrent model can correctly store and recall a long sequence of arbitrary information. In our experiment, the networks are given randomly generated eight-bit binary sequences with its length ranging from 1 to 50, followed by a finish indicator. Then the networks are asked to output the same sequence with the input.

### C.1.2 REPEAT COPY

The repeat copy task is a repeat version of the copy task. The main motivation was to see if a recurrent model could memorize input information and repeatedly reuse them at later time-steps. In

our experiment, the networks are asked to repeatedly copy the randomly generated input sequence with length ranging from 1 to 10 to output for 1 to 10 times.

### C.1.3 ASSOCIATIVE RECALL

This task test whether a memory network has the ability to retrieve memory data indirectly. The networks are given a sequence of items and then a query with one of the items, and are asked to output the subsequent item. In our experiments, each item consists of three six-bit binary vectors, and the number of items are ranging from 2 to 6.

### C.1.4 PRIORITY SORT

This task test whether a recurrent model can do priority sort——a classic elementary algorithm. This task is more challenging than previous ones as it not only require memorizing the input sequences but also making efficient use of the mutual relations of the items in the input sequences. In our experiments, the networks are given 40 eight-bit random binary keys with scalar priority values in a sequence, and are asked to output the top 30 keys in descending order of priority.

## C.2 CHARACTER-LEVEL LANGUAGE MODELLING ON PENN TREENBANK DATASET

### C.2.1 PENN TREEBANK DATASET

We use the train/validation/test split outlined in Mikolov et al. (2012). The train/validation/test batch size are 128/64/1 respectively. Following prior works on this task (Ha et al., 2016; Suarez, 2017; Mujika et al., 2017), the model is optimized over the cross-entropy loss between the predictions and the training labels, and evaluated using a bits-per-character measure. We use a single ARMIN cell of different hidden sizes, with a fully connected layer to output the predictive probabilities. The input embedding size $d_i$ is set to 128. The number of memory slots $n_{mem}$ is 20. The zoneout probability is 0.3. We train the networks for 200 epochs with a learning rate of 0.002, and decay with a factor of 10 at the last 20 epochs. We clip the gradients to a maximum norm of 1.0.

### C.2.2 HUTTER PRIZE WIKIPEDIA DATASET

We use the train/validation/test split outlined in Chung et al. (2015). The train/validation/test batch size are 128/64/1 respectively, and the vocabulary size is 205. The zoneout and dropout probabilities are 0.3 and 0.2, respectively. The input embedding size is 256. Our ARMIN network has 2 layer and 800 units at each layer, connected by a linear layer to convert 1600-dimensional features from layer 1 to 256 dimension as input to layer 2. $n_{mem}$ for each layer is set to 10. We train the networks for 50 epochs with a learning rate of 0.001, and decay with a factor of 10 at the last 10 peochs. We clip the gradients to a maximum norm of 1.0.

## C.3 TEMPORAL ACTION DETECTION AND PROPOSALS ON THUMOS' 14

The SST takes in sequential inputs of a video and outputs the existing probability predictions of action proposals with different time lengths that end at each time-step, and uses a multi-label loss to optimize its parameters. The sequential inputs are 4096-dimensional features extracted from every 16 consecutive video frames using C3D network (Tran et al., 2015), followed by PCA to downsize the input feature dimensionality.

Our experiment setup is basically the same as the SST paper, except that we use a simple fully connected layer to downsize the input features instead of using PCA. The input size of the two networks are both set to 256. The MA-SST have 256 hidden units, while the SST with LSTM have 512 hidden units. This ensures the 2 networks have roughly the same parameter count. The dropout and zoneout probabilities are both set to 0.2 and 0.3. $n_{mem}$ of the MA-SST is set to 35. We train the networks for 300 epochs with initial learning rate of 0.002, and decay with a factor of 10 at the last 60 epochs.

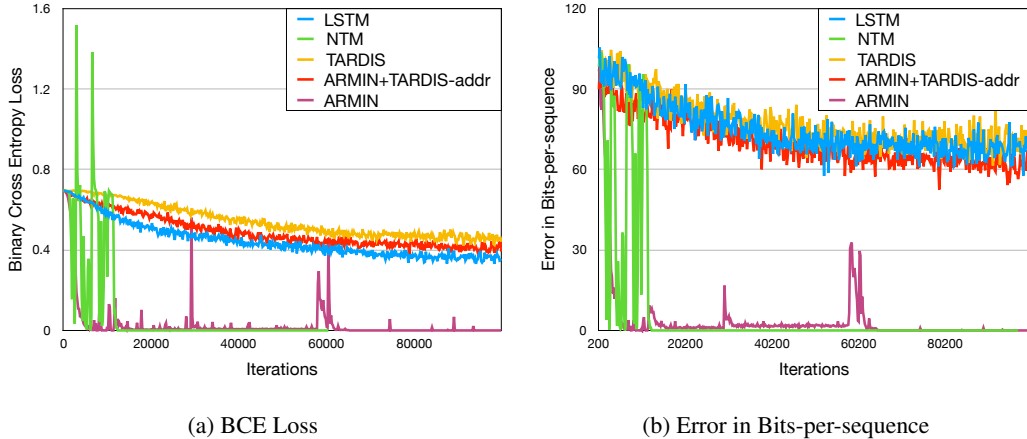

(a) BCE Loss

(b) Error in Bits-per-sequence

Figure 4: Copy task.

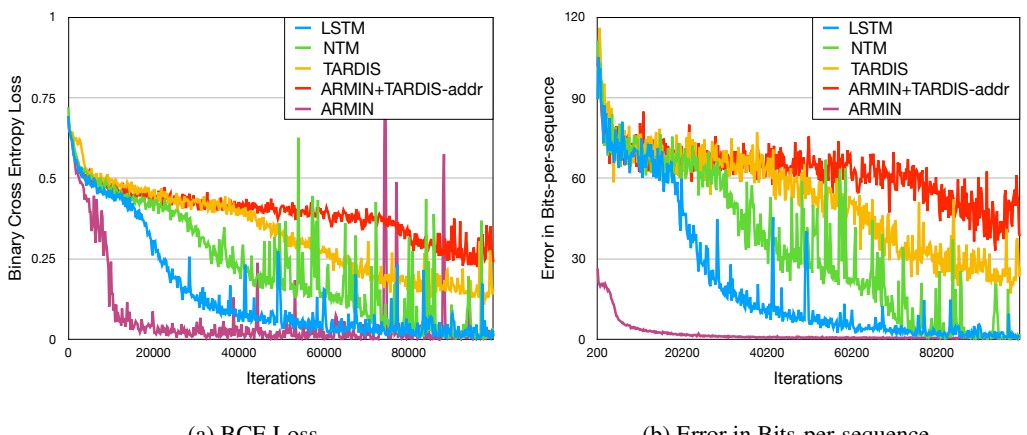

(a) BCE Loss

(b) Error in Bits-per-sequence

Figure 5: Repeat copy task.

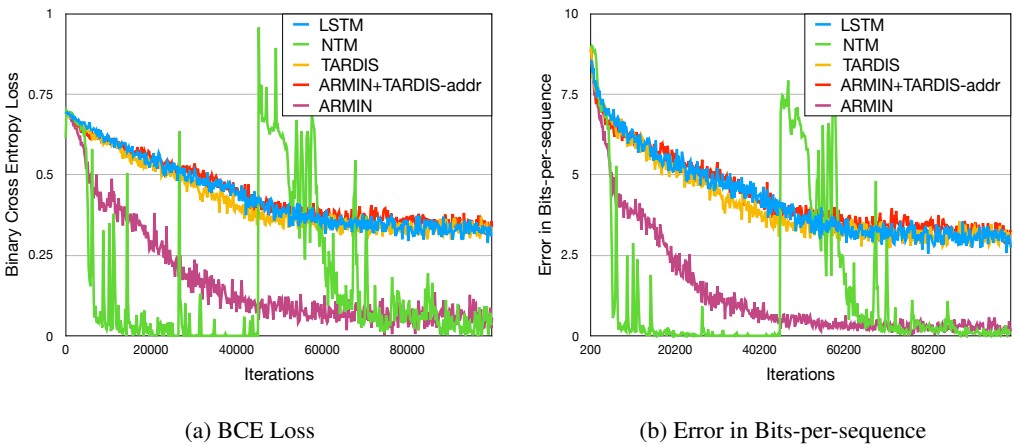

(a) BCE Loss

(b) Error in Bits-per-sequence

Figure 6: Associative recall task.

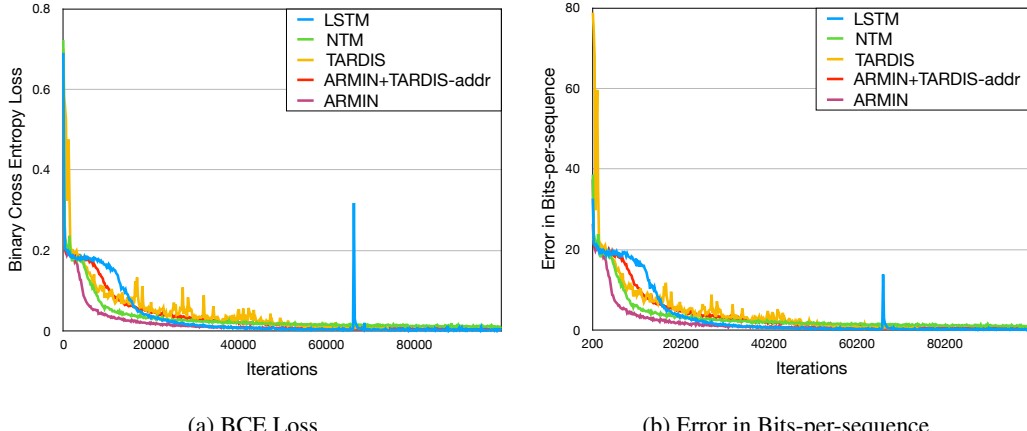

(a) BCE Loss                    (b) Error in Bits-per-sequence

Figure 7: Priority sort task.

Table 7: Ablation on Penn Treebank ($d_h$=800).

| Abaltion Model | BPC |
|---|---|
| No Control Gates | 1.354 |
| LSTM (TBPTT length=50) | 1.390 |
| ARMIN (TBPTT length=50) | 1.220 |
| Random Read | 1.224 |
| Queue-style Write | 1.352 |
| Independent Read/write | 1.245 |
| Smooth Read/write | 1.212 |

## APPENDIX D  TRAINING CURVES ON ALGORITHMIC TASKS

## APPENDIX E  ABLATION STUDY ON MEMORY AND ADDRESSING MECHANISM

In this subsection, we show the efficacy of our external memory and addressing mechanism by ablation experiments. We choose the char-level language modelling task to do the ablation research as it is a relatively standard benchmark task and has appropriate medium scale. We use the aforementioned experiment setup. To validate the efficacy of our external memory, we conduct the experiments as follows and the results are depicted in table 7:

• **Short TBPTT Length** We shorten the truncated length in TBPTT from 150 to 50. The gradients are truncated so we can't do backpropagation on hidden states or memory from last iteration, but the ARMIN still has direct access to historical hidden states. By comparing the performance of the LSTM and ARMIN under such circumstances, we can know the efficacy of the external memory. The results show that the LSTM has a dramatic performance drop of 1.390 BPC while the ARMIN only has a minor performance drop of 1.220 BPC. The results confirm that the direct access to memory is crucial when the networks can't access historical information by backpropagating gradients. And notably, our experiment shows that the ARMIN is very robust to short-length TBPTT, $i.e.$ small iteration length, which means lower memory consumption. It enables using very large batch size to speed up the training and further increasing performance on sequence modelling tasks.

• **No Control Gates** We remove the control gate $\boldsymbol{g}_t^h$ and $\boldsymbol{g}_t^r$ in ARMIN cell as depicted in equation 3, and directly use $\boldsymbol{x}_t$ and $\boldsymbol{h}_{t-1}$ to compute the gates in equation 6. The result shows a BPC of 1.354 that has a dramatic performance drop compared the 1.202 BPC of normal ARMIN. From this, we deduce that the control gates successfully filter out irrelevant information. Without the control gates, the historical state $\boldsymbol{r}_t$ would have bring more noise than useful information.

To validate the efficacy of the addressing mechanism of the ARMIN, we do the following modifications to the read/write operations of the ARMIN, and the results are also depicted in table 7:

• **Random Read** The ARMIN randomly chooses a historical state while reading. The result shows a BPC of 1.224, compared to the 1.202 BPC of normal ARMIN. It proves that the read regime we proposed in section 3.2 is effective and can choose more useful historical state than random read.

• **Queue-style Write** While writing, the ARMIN follows a fashion of first-in-first-out, $i.e.$ always chooses the oldest memory slot to write. It makes the memory $M$ stubbornly store the hidden states of the last 20 time-steps. The result shows a BPC of 1.352, which proves our normal read/write mechanism has much more flexibility and can learn long-term patterns.

• **Independent Read/write** While writing, the ARMIN doesn't write to the last read memory slot, but independently choose a memory slot to overwrite according to $x_t$ and $h_t$, in a similar fashion with the read operation we presented in section 3.2. The result shows a BPC of 1.245, proving that the normal writing regime in the ARMIN can better learn long-term patterns and maximally preserve historical information, as we explain in section 3.3.

• **Smooth Read/write** We replace the gumbel-softmax function with the vanilla $\mathrm{softmax}$ function, so the ARMIN reads and writes in a smooth way. The results show a minor performance drop of 1.212 BPC. The result shows that our discrete read/write regime is still better. Moreover, the smooth read/write regime would consume much more memory at both training and inference stages.

