# OpenReview forum: "SEQUENCE MODELLING WITH AUTO-ADDRESSING AND RECURRENT MEMORY INTEGRATING NETWORKS"
_ICLR.cc/2019/Conference_

### Official Review · AnonReviewer1 · 2018-11-01
**Review for "Sequence Modelling with Memory-Augmented Recurrent Neural Networks"**

**Rating:** 5
**Confidence:** 4

**Review:**

The paper proposed a RNN with skip-connection (external memory) to past hidden states, this is a slightly different version of the TARDIS network. The authors experimented on PTB and a temporal action detection method.

Novelty:

I dont see a lot of novelty to the method. The authors proposed a method very similar to TARDIS, the difference seems to be that MMARNN does not use extra usage vectors for reading from previous memory, but this is not a fundamental difference between MMARNN and Tardis.

Shortcomings of the paper:

1. The experiments seem rather weak. The authors experimented on PTB and temporal action detection method. It is not clear why authors experimented with PTB, this is not a task with long-term dependencies, I do not see how this task (compared to many other tasks) can benefit from using external memory (especially when only 1 past hidden state is used

2. The model uses a single past hidden state, it is not clear to me why this is better than using  a weighted sum of a few past hidden states, as many tasks requires long-term dependencies from multiple steps in the past. The authors should cite "Sparse attentive backtracking" (https://arxiv.org/abs/1809.03702) at NIPS 2018. SAB is very related in that it also propagate gradients to a few hidden states in the memory. The difference is that SAB used a few hidden states from the past/ memory instead of one; another difference is that it propagates gradients locally to the selected hidden states/ memory slots.

3. The paper only demonstrated experimental results on PTB and temporal action prediction. I think it would make the paper a lot stronger if the authors experimented with a variety of  different tasks. Tasks that requires long term dependencies can really demonstrate the strength of the model (copy and adding tasks).

4. If the authors could run the model on copy and adding tasks, I would be curious to see if the model is picking the "correct" timestep in the memory / past.

post rebuttal: I feel that the authors have addressed some of my concerns, in particular, in terms of additional experimental results. I have raised the score to reflect this changes.

---

> ### Author Response · Authors · 2018-11-26
> **Clarification and more benchmark results are reported.**
>
> Thank you for your valuable comment. We apologize for not clearly stating our difference against TARDIS and causing some misunderstanding regarding the novelty. We have accordingly revised our paper and can address your concerns as follows:
>
> Q1:Novelty:I don't see a lot of novelty to the method. The authors proposed a method very similar to TARDIS, the difference seems to be that MMARNN does not use extra usage vectors for reading from previous memory, but this is not a fundamental difference between MARNN and Tardis.
>
> We have discussed 5 important differences regarding the cell computation and addressing mechanism , supported by experimental evidences we added. Please refer to appendix B for a thorough comparison.
>
> Q2: The experiments seem rather weak. The authors experimented on PTB and temporal action detection method. It is not clear why authors experimented with PTB, this is not a task with long-term dependencies, I do not see how this task (compared to many other tasks) can benefit from using external memory (especially when only 1 past hidden state is used.
>
> We apologize for not clearly stating the motivation of our dataset choice of PTB and THUMOS 14. As we state in the introduction, we aim at building a bridge between simple RNN and complex memory models. We primarily intend to focus on augmenting RNN's performance on real-world tasks with a light-weight external memory, but not focus on coming up with more functional and complex memory mechanism.  We choose PTB and THUMOS 14 tasks  because they are real-world tasks where we can examine if our memory network can be helpful.  In our network, each r_t , although only 1 past hidden state, is a summary of historical hidden states with skip connection selected by auto-addressing mechanism. The ARMIN(renamed from MARNN) cell learns to recurrently integrate the summary of historical information from r_t into h_t at every time-step, which favors a paradigm of deep transition, as is shown by the success of  deep-transition RNNs on PTB task.  We believe this  can explain why we choose PTB and why the ARMIN success on PTB.
>
> Q3: The model uses a single past hidden state, it is not clear to me why this is better than using  a weighted sum of a few past hidden states, as many tasks requires long-term dependencies from multiple steps in the past. The authors should cite "Sparse attentive backtracking" (https://arxiv.org/abs/1809.03702) at NIPS 2018. SAB is very related in that it also propagate gradients to a few hidden states in the memory. The difference is that SAB used a few hidden states from the past/ memory instead of one; another difference is that it propagates gradients locally to the selected hidden states/ memory slots.
>
> Thank you for pointing out this paper, and we have cited and introduced the SAB in our paper. As we state in Q1, each r_t is a summary of historical hidden states with skip connection selected by auto-addressing mechanism(we train the selecting process by backpropagation), so there is no fundamental difference from using  a weighted sum of a few past hidden states.  Another reason is that the gumbel-softmax can only sample 1-hot vector, so we think a more fundamental comparison between ARMIN and SAB should be : is the 1-hot sampling of gumbel-softmax better or multiple selecting of ReLU and softmax addressing mechanism in SAB is better? Due to time constraints, we leave this for future work.
>
> Q3&4. The paper only demonstrated experimental results on PTB and temporal action prediction. I think it would make the paper a lot stronger if the authors experimented with a variety of  different tasks. Tasks that requires long term dependencies can really demonstrate the strength of the model (copy and adding tasks). If the authors could run the model on copy and adding tasks, I would be curious to see if the model is picking the "correct" timestep in the memory / past.
>
> We have added a set of algorithmic tasks introduced by the NTM paper, including copy, repeat copy, associative recall and priority sort. We think these tasks do require exact long-term dependency, and we demonstrate the efficiency of our network on these tasks. Specifically, our network can converge 3~4 time faster in terms wall clock time than NTM on most of the tasks. We also adds enwik8 char-level lm tasks. We are currently doing experiment on pMNIST, but can't report in time as the training is very slow. We will report the results in later revision of our paper.
>
> We hope these answers can address your concerns, thanks!

---

> > ### Comment · AnonReviewer1 · 2018-12-13
> > **Post-rebuttal reviews**
> >
> > I feel that the authors have addressed some of my concerns in the reviews.
> >
> > In particular, authors added additional results  of the model performance on various different tasks including copying, pMNIST that may entail long-term dependency.
> >
> > I am willing to raise the scores of my review.

---

### Official Review · AnonReviewer2 · 2018-11-02
**Efficiency of MARNN compared to other models?**

**Rating:** 4
**Confidence:** 4

**Review:**

This paper introduces a memory-augmented RNN (MARNN) which aims at being lightweight and   differentiable. In a nutshell, authors propose to augment a LSTM-type architecture with several memory cells. At each time-step, MARNN retrieves one memory cell, updates his state, and updates the memory cell content. To learn the retrieval operation that requires discrete addressing,  authors rely on the Gumbel-Softmax. Authors evaluate their approach on PennTreeBank character level modelling where they demonstrate competitive performances. They also report state-of-art performance on the Thumos dataset. The paper is overall clear and pleasant to read.

Authors highlight that MARNN is more lightweight compared to existing memory networks. MARNN can indeed retrieves only memory cells at inference. However,  since MARNN uses a Gumbel-Softmax to train the discrete addressing scheme, it is it not clear if there is any advantage in term of memory and computation of MARNN relatively to other network during training? It would be nice to compare the computation time/memory usage of MARNN with other memory augmented network such as TARDIS, NTM or Memory Network during training and inference.

Another claim is that MARNN can possibly boost training speed by reducing the lengths of TBTT.  But MARNN also haves a training time overhead as showed in Figure 2.  How does the overall training time/performances of MARNN with TBPTT of 50 compared to a LSTM with TBPTT of 100/150?

The writing can be sometime a bit imprecise. For instance authors say that MARNN “learns better representations that many hierarchical RNN structure”. I agree that MARNN outperforms in term of accuracy, however, it is not clear what the author are referring to by “better representation” of the MARNN hidden state? Performance gain of MARNN could also be due to the external memory which allows  to retain more information of the input? In addition, it would be nice to precise which type of hierarchical RNN structure MARNN does (or doesn’t) outperform. Another claim is that MARNN can “easily learn long-term dependencies”. While this is reasonable, I am unsure that the empirical evaluation support this.  It would be nice to show how the gradients backpropagated through time behave in practice to support this claim?


Memory-augmented network are a very important research directions and the MARNN architecture is interesting. However, it is not entirely clear to me what is the main advantage of MARNN relatively to other memory networks network such as TARDIS, NTM or Memory Network for training and/or inference. Although authors do compare with TARDIS, further comparison with the other networks and in term of computation time and memory could help clarify those points.

---

> ### Author Response · Authors · 2018-11-26
> **Concerns mostly addressed.**
>
> Thank you for your valuable comment and pointing out the insufficient evidence in our work to support our claims. We have added more experiments and can address your concerns as follows:
>
> Q1:It would be nice to compare the computation time/memory usage of MARNN with other memory augmented network such as TARDIS, NTM or Memory Network during training and inference.
>
> We have compared our network (renamed as ARMIN)  with LSTM, NTM, TARDIS, ARMIN+TARDIS-addr in terms of speed/memory consumption during training and inference in section 5. The results shows our network constantly outperforms other memory networks.
>
> Q2:Another claim is that MARNN can possibly boost training speed by reducing the lengths of TBTT.  But MARNN also haves a training time overhead as showed in Figure 2.  How does the overall training time/performances of MARNN with TBPTT of 50 compared to a LSTM with TBPTT of 100/150?
>
> We conduct a control experiment in section 5, and compare 2 different setup of ARMIN with TBPTT  of 50 against LSTM with TBPTT of 100/150. We show that ARMIN setup 1 obtain 40% training speed gain than LSTM with TBPTT of 150 while keeping a slightly better performance and 16% less parameter count; in ARMIN setup 2, we observe that with only 8.8% more memory consumption, we obtain about 1.5 points of BPC performance gain and 7.1% training speed gain under similar parameter count.
>
> Q3:The writing can be sometime a bit imprecise. For instance authors say that MARNN “learns better representations that many hierarchical RNN structure”. I agree that MARNN outperforms in term of accuracy, however, it is not clear what the author are referring to by “better representation” of the MARNN hidden state? Performance gain of MARNN could also be due to the external memory which allows  to retain more information of the input?
>
> By saying “better representations”, we refer to the concatenation of the gated contents from h_t and r_t.  The performance gain comes from two aspect. a)If we remove the gated contents of r_t from o_t, the ARMIN undergoes a BPC performance drop from 1.202 to 1.220, which is still better than the best performing BPC of 1.24 of the LSTM. b)The rest of the performance gain comes from the recurrent memory integration of the ARMIN cell, which also favors a paradigm of deep transition, as is shown by the success of the deep transition RNNs on this task. Please refer to line 10 in page 7 for more details.
>
> Q4:It would be nice to precise which type of hierarchical RNN structure MARNN does (or doesn’t) outperform.
>
> We have specified in our paper that our network outperforms HM-LSTM, 2-Layer HyperLSTM and 21 layer IndRNN, and is outperformed by state-of-the-art BPC of 1.19 of HyperRHN and FS-LSTM.
>
> Q5:Another claim is that MARNN can “easily learn long-term dependencies”. While this is reasonable, I am unsure that the empirical evaluation support this.
>
> We add algorithmic tasks which require exact long-term dependencies introduce by the NTM paper, where we demonstrate the fast converge speed in most of the tasks both in terms of wall clock time and iteration numbers. We think this can support our claim of being able to “easily learn long-term dependencies”.
>
> Q6:it is not entirely clear to me what is the main advantage of MARNN relatively to other memory networks network such as TARDIS, NTM or Memory Network for training and/or inference. Although authors do compare with TARDIS, further comparison with the other networks and in term of computation time and memory could help clarify those points.
>
> We have done further comparison to address the concern, as we state in Q1. Our main advantage with respect to NTM and TARDIS are faster running speed, less memory consumption and better performance in real-world tasks (where we doubt the practicability of the carefully designed and complex addressing mechanism of the NTM).
>
> We hope these answers can address your concerns, thanks!

---

### Official Review · AnonReviewer3 · 2018-11-08
**Differentiate from TARDIS and show more benchmark results.**

**Rating:** 4
**Confidence:** 5

**Review:**


Summary:

This paper introduces a new RNN architecture with external memory for sequence modeling. The proposed architecture (MARNN) is a simplification of TARDIS (Gulcehre et al., 2017). It uses the similar reader-writer tying mechanism, gates to control information flow from previous hidden state and memory. However, it has a simpler addressing mechanism. Authors show results in Character level PTB and a temporal action detection/proposal task.

Major comments:

MARNN looks like a simplification of TARDIS architecture with Gumbel softmax. The major difference between the two architectures is the addressing mechanism.

1.	Can the authors clearly differentiate MARNN vs TARDIS?

2.	Authors compare against TARDIS only in character level PTB which is actually a task which does not require very long term dependencies. It would be better if authors consider more tasks and directly compare against the TARDIS addressing mechanism to prove that the proposed addressing mechanism is indeed better.

3.	Authors should consider more tasks, to show the efficiency of the proposed architecture.

4.	The name of the model seems to be too generic. NTM, TARDIS, DNC with recurrent controller can be considered as memory augmented RNN. Please change the name.

---

> ### Author Response · Authors · 2018-11-26
> **Differences with TARDIS and more benchmark results are reported.**
>
> Thank you for your valuable comments and pointing out the existing problems of our work. We agree that the MARNN name seems to be too generic and have renamed it to "ARMIN" network. We address your other comment as follows:
>
> Q1. Can the authors clearly differentiate MARNN vs TARDIS?
>
> We have clearly differentiate ARMIN(MARNN) against TARDIS with 5 important differences in appendix B. The main differences not only comes from the addressing mechanism but also from the cell computations. We have respectively validated the efficiency of the two aspect of differences in algorithmic tasks and char-level language modelling tasks.
>
> Q2. Authors compare against TARDIS only in character level PTB which is actually a task which does not require very long term dependencies. It would be better if authors consider more tasks and directly compare against the TARDIS addressing mechanism to prove that the proposed addressing mechanism is indeed better.
>
> We have directly compared ARMIN against TARDIS and ARMIN+TARDIS-addr in experiments and validates the efficiency of ARMIN as we states in Q1.
> We also apologize for not clearly stating the motivation of our dataset choice of PTB and THUMOS 14. As we state in the introduction, we aim at building a bridge between simple RNN and complex memory models. We primarily intend to focus on augmenting RNN's performance on real-world tasks with a light-weight external memory, but not focus on coming up with more functional and complex memory mechanism.  We show in our experiments that our network  is effective in more real-world tasks(PTB and THUMOS14) and is more light-weight than other memory networks.
>
> Q3. Authors should consider more tasks, to show the efficiency of the proposed architecture.
>
> We have added a set of algorithmic tasks introduced by the Neural Turing machine paper, as well as the char-level language modelling task on enwik8 dataset. We are also doing experiments on pMNIST task but we can't report the results in time cause the pMNIST experiments runs very slow. We will report the results in later revisions.
>
> We hope these answers can address your concerns, thanks!

---

### Author Response · Authors · 2018-11-26
**Revision**

We thank the reviewers for their valuable suggestions and agree on most of the opinions proposed by the reviewers. We have added some experiments and accordingly revised our paper.

For a brief overview:
1.Following the suggestion of Reviewer3, we have renamed our model from "MARNN" to "Auto-addressing and Recurrent Memory Integration Network(ARMIN)".
2.We have added a set of algorithmic tasks introduced by the Neural Turing machine paper, as well as the char-level language modelling task on enwik8 dataset. We add comparison with NTM, TARDIS and ARMIN+TARDIS-addr( ARMIN cell with TARDIS addressing mechanism) in these experiments, and demonstrate significant advantage over these networks.
3.We have tested the speed and memory advantage of our network over NTM and TARDIS in section 5.We also present a ARMIN setup that runs 40% faster than LSTM while keeping similar performance.
4.We have clearly differentiate our network from TARDIS with 5 important differences in appendix B.
5.We have moved all implementation details of experiments from the main body to appendix C.
6.We have moved our ablation study from the main body to appendix E.
7.Some statement revision for better consistency.

We hope our revision can address most of the concerns proposed by the reviewers.

---

> ### Author Response · Authors · 2018-12-05
> **Update: pMNIST test result added**
>
> We have tested our model against TARDIS and LSTM in the pMNIST task, we believe this is a task where the networks must integrate and remember observed information from previous pixels,and thus, can test the efficiency of the recurrent memory integration of ARMIN.
> We use the Adam optimizer to train the models without layer-norm  and zoneout for 150 epochs, and we get the following results:
>
> ---------------------------------------------------------------------------------------------------------------------------------------
> model    | Acc(%) |   Hidden Size  | memory Size |Param Count  |  Memory (GB) |Time(min/epoch)
>  ---------------------------------------------------------------------------------------------------------------------------------------
> LSTM      | 92.7      |        128           |            --           |          69k         |   1.687               |           7
> NTM       | 92.5      |        100           |        28x28       |           68k        |    2.912              |           40
> TARDIS   | 94.1      |        100           |        28x28       |          74k         |    6.919              |           24
> ARMIN   | 94.3      |        100           |        28x28       |          81k         |    2.713              |           15
>
> We will add this experiment to later revisions. We will also opensource our code of all our experiments with reproducibility.
>
> #update: NTM result added.

---

### Meta-Review · Area_Chair1 · 2018-12-14
**a bar is higher for a new memory augmented neural network**

**Confidence:** 4
**Recommendation:** Reject

**Metareview:**

there have been many variants of memory augmented neural nets since around 2014 when NTM, attention-based NMT and MemNet were proposed. it is indeed still an interesting and important direction of research, but the bar for introducing yet another variant of memory-augmented neural nets has been significantly raised, which is a sentiment shared by the reviewers. the author's response had not swayed the reviewers' opinion, and i am sticking to the reviewers' decisions.

i believe more streamlined and systematic comparison among different memory augmented networks across many different benchmarks (e.g., use the same set of latest variants of memory nets across all the benchmarks) in this submission would make it a better paper and increase the chance of acceptance.